# Could Small Language Models Serve as Recommenders? Towards Data-centric Cold-start Recommendations

## ABSTRACT

Recommendation systems help users find information that matches their interests based on their historical behaviors. However, generating personalized recommendations becomes challenging in the absence of historical user-item interactions, a practical problem for startups known as the *system cold-start recommendation.* Current research tackles user or item cold-start scenarios but lacks solutions for system cold-start. To tackle the problem, we initially propose PromptRec, a simple but effective approach based on in-context learning of language models, where we transform the recommendation task into the sentiment analysis task of natural languages containing user and item profiles. However, this naïve strategy heavily relied on the strong in-context learning ability emerged from *large* language models, which could suffer from significant latency for online recommendations. To fill this gap, we present a theoretical framework to formalize the connection between in-context recommendation and language modeling. Based on it, we propose to enhance *small* language models with a data-centric pipeline, which consists of: (1) constructing a refined corpus for model pre-training; (2) constructing a decomposed prompt template via prompt pre-training. They correspond to the development of training data and inference data, respectively. To evaluate our proposed method, we introduce a cold-start recommendation benchmark, and the results demonstrate that the enhanced small language models can achieve comparable cold-start recommendation performance to that of large models with only around 17% of their inference time. To the best of our knowledge, this is the first study to tackle the system cold-start recommendation problem. We believe our findings will provide valuable insights for future works. The benchmark and implementation of the methods are available at https://anonymous.4open.science/r/PromptRec-C3EF/.

## KEYWORDS

In-context learning, cold-start recommendation, language models, data-centric AI.

**ACM Reference Format:**
Anonymous Author(s). 2018. Could Small Language Models Serve as Recommenders? Towards Data-centric Cold-start Recommendations. In *Proceedings of Make sure to enter the correct conference title from your rights confirmation emai (Conference acronym 'XX).* ACM, New York, NY, USA, 10 pages. https://doi.org/XXXXXXX.XXXXXXX

## 1 INTRODUCTION

Recommendation systems filter massive online information and help users discover items tailored to their interests. Traditional recommendation systems such as collaborative filtering [19, 20, 53] and content-based methods [11] rely on historical user-item interactions (e.g., clicks, purchases, ratings) to learn user/item representations and find matched items for users. However, this pipeline would fail in scenarios where we could *not* obtain any user-item interactions, and we call it the *system cold-start recommendation* problem, which typically happens in situations such as start-up businesses [3, 40]. Although cold-start recommendation scenarios have been studied in previous research [23, 33, 39], as illustrated in Figure 1, they still assume historical user-item interactions are available for training or during inference, which differs from our setting. A straightforward strategy to tackle the system cold-start recommendation problem is to manually design rules, such as recommending popular or seasonal items [8, 39], but the recommendation results are less personalized and could hurt user experiences.

One potential direction to solve the system cold-start recommendation problem is leveraging in-context learning with large language models (LLMs) [4, 30, 52], demonstrated by LLMs could quickly adapt to a new task (recommendation in our study) without training them on a task dataset. Intuitively, the relation between user interests and item properties has been implicitly expressed as a natural language in public corpora. Thus, it could be captured by LLMs and used for cold-start recommendations. However, in-context learning is an emergent ability of large language models, which usually starts from hundreds of millions to billions of parameters. Therefore, adopting these methods for online recommendation is impractical due to their slow and costly inference. This naturally raises a question: **could *small* language models be in-context recommenders for system cold-start recommendation?**

To answer this question, we initially propose a simple but effective in-context learning approach by leveraging language models, named *PromptRec*, to tackle the system cold-start recommendation problem. Specifically, PromptRec first maps profile features of users and items to natural language descriptions, then applies a template reformatting the recommendation task as a language modeling task over binary sentiment words, and finally leverages a language model to accomplish the task and perform recommendations. Our pilot experiments show that **large language models successfully makes personalized cold-start recommendations, but small language models fail**, consistent with the scaling law [22, 59] observed in the emergent in-context learning ability of LLMs.

To analyze the reason behind this failure, we propose a theoretical framework to formalize the mechanism of in-context recommendation with PromptRec under the Hidden Markov Model (HMM) assumption [61]. Under the assumption, a language model first infers the "concept" (sentiment polarity in this study) based on the input prompt (user-item profiles and a template in this study), and then makes recommendations conditioned on both the inferred

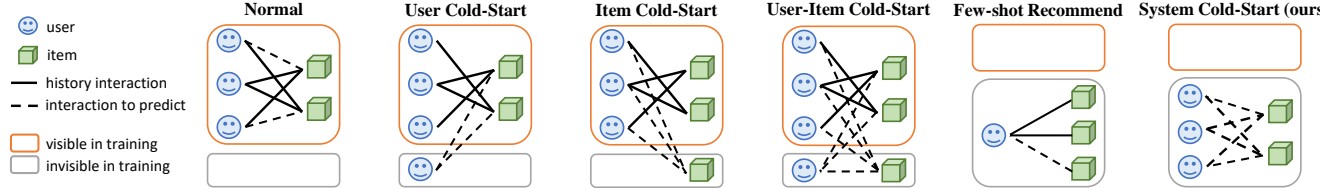

**Figure 1: Illustration of different cold-start recommendation scenarios, including user cold-start [39], item cold-start [33], user-item cold-start [23], few-shot recommend [5, 66], and system cold-start (ours).**

concept and the input prompt. Our analysis demonstrates that language models perform in-context recommendations by estimating the likelihood of the sentiment words conditioned on the user-item context and the prior probability to different pre-trained concepts. Small language models fail to provide precise estimations of these two factors due to their limited parameters.

With this finding, we enhance the small language models' recommendation performance with improved estimation of in-context prediction probabilities via a data-centric pipeline, including two major steps: model pre-training and prompt template pre-training. Specifically, if we could (1) pre-train the language model on a corpus related to the recommendation scenario, and (2) pre-train the prompt template with interactions from other recommendation domains, the cold-start recommendation performance of small language models could be enhanced. Note that the training data belongs to distributions different from the target recommendation domain, so it does not violate the system cold-start setting. However, we face two challenges due to the lack of user-item interactions in the cold-start scenario: (1) how to find a corpus to pre-train small models for recommendation tasks? (2) how to make the prompt template generalizable across different domains? For the first challenge, we propose refining a general corpus by maximizing mutual information between the general documents and the synthetic user-item interactions in the cold-start scenario. For the second challenge, we propose to decompose prompt templates into the "task" and "domain" prompts, where the former is transferable across different recommendation scenarios. We verify these two methods and show that **the enhanced *small* language models achieve comparable performance with large models in just around** 17% **of their inference times**. Remarkably, enhanced BERT-mini with 11.3M parameters (≈3% of BERT-large) achieve performance comparable with BERT-large in terms of in-context recommendations under the system cold-start setting. We summarize contributions as follows:

- We formalize the system cold-start recommendation problem and introduce the first benchmark to our community.
- We propose PromptRec for system cold-start recommendation and provide the first theoretical framework to formalize the in-context recommendation with language models.
- We explore the potential of using small language models by leveraging general corpus and cross domain datasets.

## 2 PRELIMINARY

### 2.1 Notations

In this work, we use boldface lowercase letters (e.g., $\mathbf{c}$) to denote vectors, boldface uppercase letters (e.g., $\mathbf{R}$) to denote matrices, and calligraphic capital letters (e.g., $\mathcal{D}$) to denote sets. Specifically, each

recommendation dataset $\mathcal{D} = (\mathcal{U}, \mathcal{I}, \mathbf{R})$ has a user set $\mathcal{U}$, an item set $\mathcal{I}$, and a matrix storing user-item interactions $\mathbf{R} \in \mathbb{R}^{|\mathcal{U}| \times |\mathcal{I}|}$, where $r_{u,i} \in \mathbf{R}$ indicates the interaction between the user $u$ and the item $i$. Each user and each item has $d_U$ and $d_I$ profile features denoted as $\mathbf{c}_u \in \mathbb{R}^{d_U}$ and $\mathbf{c}_i \in \mathbb{R}^{d_I}$, respectively. The profile features are attributes that describe users or items (e.g., user's age, gender, and occupation; item's name, brand, and category).

### 2.2 Problem Statement

We choose the click-through rate (CTR) prediction task [41] to set up our recommendation scenario. That is, each record $r_{u,i} \in \{0, 1\}$ is a binary value, where $r_{u,i} = 1$ means user $u$ clicked item $i$. The recommendation system $f$ takes a user-item pair $(\mathbf{c}_u, \mathbf{c}_i)$ as *input* to predict the probability $\hat{r}_{u,i} \in [0, 1]$ that the user will click on the item as *output* based on model $\hat{r}_{u,i} = f(r_{u,i} = 1 | \mathbf{c}_u, \mathbf{c}_i)$. The goal of CTR prediction is to minimize the difference $\mathcal{L}$ between the predicted probability $\hat{r}_{u,i}$ and the real user-item interaction $r_{u,i}$.

### 2.3 System Cold-Start Recommendation

Under the system cold-start recommendation setting, we can *not* obtain any interaction records, which is a common situation for start-up companies that have just launched their businesses [40]. Therefore, in the system cold-start recommendation, we define a *target dataset* as $\mathcal{D}_{\text{tgt}} = (\mathcal{U}_{\text{tgt}}, \mathcal{I}_{\text{tgt}}, \mathbf{R}_{\text{tgt}})$ with an empty interaction matrix $\mathbf{R}_{\text{tgt}} = \emptyset$. Our goal is to recommend items in $\mathcal{I}_{\text{tgt}}$ to users in $\mathcal{U}_{\text{tgt}}$ by using their profile features $\{\mathbf{c}_u\}$ and $\{\mathbf{c}_i\}$. We do not allow the use of recorded interactions, no matter in training or in inference, but recommendation system developers could explore available resources to build user and item profiles.

## 3 METHODOLOGY

We introduce the details of the proposed framework. In Sec. 3.1, we present PromptRec, an in-context learning approach to tackle the system cold-start problem. In Sec. 3.2, we provide a theoretical framework to build the connection between in-context recommendation and language model. Furthermore, we develop a data-centric pipeline to enhance the in-context recommendation of (small) language models, with a focus on training corpus refinement in Sec. 3.3, and inference-time prompt design in Sec. 3.4.

### 3.1 PromptRec

The traditional supervised learning paradigm fails under the system cold-start setting because there is no training data for tuning the model $f$. The emergent in-context learning ability [4, 46] of LLMs is a potential way to overcome this challenge, where the downstream task is formatted as one of the language model pre-training tasks,

also called "prompt" learning [29]. Recent prompt-based recommendation systems [5, 47, 66] usually align the recommendation process with the language modeling task, where LLMs are adopted to estimate the probability of the item's name appearing within a user-item context. For example, given a context about the user interactions as "*A user clicked hiking shoes, will also click trekking poles*", they treat the probability that "trekking poles" appears within this context as the user preference to the candidate item "trekking poles". However, simply predicting item names is ineffective for recommendation, especially in the zero-shot situation. This is because the probability of the item's name is affected by every single word in it, where an item with a name made up of common words will have a higher chance to appear, regardless of the context. For example, "League of Legends" naturally has a higher probability than "Legend of Zelda" since "League" is more common than "Zelda" in corpora.

To solve this problem, instead of predicting the item names, we give recommendations by predicting the probability of chosen binary words. In practice, these words can be sentiment words (e.g., "good", "bad"). Predicting the probabilities associated with sentiment words can offer a more accurate representation of user preferences by mitigating the influence of rare or high-frequency words on the final recommendation outcomes. Formally, we define a prompting function $f_{\text{prompt}}$ that maps the user-item pair $(\mathbf{c}_u, \mathbf{c}_i)$ into a context $X_{u,i} = f_{\text{prompt}}(c_u, c_i)$. By giving a language model $f_{\text{LM}}$, the preference score $\hat{r}_{u,i}$ from user $u$ to item $i$ is estimated by:

$$\hat{r}_{u,i} = \frac{P(\mathcal{V}_{\text{pos}})}{P(\mathcal{V}_{\text{pos}}) + P(\mathcal{V}_{\text{neg}})}, \tag{1}$$

where

$$P(\mathcal{V}') = \frac{1}{|\mathcal{V}'|} \sum_{w \in \mathcal{V}'} \log p(w|X_{u,i}). \tag{2}$$

Here, $\mathcal{V}_{\text{pos}}, \mathcal{V}_{\text{neg}} \subset \mathcal{V}$ are the predefined positive and negative sentiment vocabulary sets, $\mathcal{V}$ is the full vocabulary set; $p(w|X_{u,i})$ is the predicted probability from the language model $f_{\text{LM}}$ that generates word $w$ conditioned on the context $X_{u,i}$ with in-context learning. We first manually design a template $\mathcal{T}$ as "*The player is a* age gender occupation. name *is categorized as a* category *video game created by* producer. *Overall, the player feels* [MASK] *about the game.*", where each underlined word is a slot. We then fill the template slots with the user and item profile features $c_u$ and $c_i$, which have been converted into natural language at first[1]. If we let $\mathcal{V}_{\text{pos}} = \{\text{"good"}\}$ and $\mathcal{V}_{\text{neg}} = \{\text{"bad"}\}$, the predicted preference $\hat{r}_{u,i}$ is computed by normalizing the probabilities of observing "good" and "bad" at position [MASK].

## 3.2 In-context Recommendation Framework

Assume we only consider one word for each sentiment vocabulary set, i.e., $|\mathcal{V}_{\text{pos}}| = |\mathcal{V}_{\text{neg}}| = 1$, then the objective of making cold-start recommendation for a user-item pair with Eq. (1) is defined as:

$$\min \mathcal{L}(r_{u,i}, \hat{r}_{u,i}) \rightarrow \max p(y_{u,i}|X_{u,i}), \tag{3}$$

where $y_{u,i}$ denotes the only positive word in $\mathcal{V}_{\text{pos}}$ if the ground-truth preference $r_{u,i} = 1$; otherwise it is the only negative word in

---

[1]In this example, the user-item context $X_{u,i}$ could be "*The player is a young male college student. Legend of Zelda is categorized as a action adventure video game created by Nintendo. Overall, the player feels* [MASK] *about the game.*"

$\mathcal{V}_{\text{neg}}$. The equation above indicates that the effectiveness of recommendation heavily relies on the accurate estimation of $p(y_{u,i}|X_{u,i})$, which is achieved by leveraging LLMs. However, a drawback of employing LLMs in online recommendation is their relatively slow inference speeds [25, 27]. One way to overcome this dilemma is adopting small language models in PromptRec, but they are recognized as having limited in-context learning abilities [4, 30, 32, 36].

To analyze how to enhance the in-context learning ability of a language model, we extend $p(y_{u,i}|X_{u,i})$ by assuming that a language model generates words as a Hidden Markov Model (HMM) suggested by [61]. Under the HMM assumption [1], a language model $f_{\text{LM}}$ generates words through a two-step process, where it first draws a concept $\theta \in \Theta$ from concept bases $\Theta$ and then samples a sequence of words conditioned on the concept. Thus, we can extend the in-context recommendation objective function as:

$$\max p(y_{u,i}|X_{u,i}) = \max \int_{\theta \in \Theta} p(y_{u,i}|X_{u,i}, \theta)p(\theta|X_{u,i})d\theta$$
$$\propto \max \int_{\theta \in \Theta} p(y_{u,i}|X_{u,i}, \theta)p(X_{u,i}|\theta)p(\theta)d\theta. \tag{4}$$

An effective recommendation model requires an accurate estimation of each factor in the above equation. In Sec. 3.3, we introduce a data refinement strategy for better probability estimation by model pre-training. In Sec. 3.4, we introduce a prompt refinement method for a better design of $X_{u,i}$ by prompt pre-training.

## 3.3 Refining Corpus for Model Pre-training

### 3.3.1 Model Pre-training Meets In-context Recommendation.
We begin by theoretically analyzing why language models pre-trained on a text corpus can be used for recommendation tasks. Specifically, a language model is pre-trained to maximize the likelihood of any observed $T$-length sequence $\mathcal{W} = [w_1, ..., w_T]$, where each word $w$ belongs to $\mathcal{V}$. Following the HMM framework, the pre-training objective is written as:

$$\max p(\mathcal{W}) = \max \int_{\theta \in \Theta} p(\mathcal{W}|\theta)p(\theta)d\theta. \tag{5}$$

This objective encourages the language model to distinguish different concepts and the word probabilities conditioned on the concepts during model pre-training. If a pair of instance $(X_{u,i}, y_{u,i}) \in \mathcal{D}_{\text{tgt}}$ is present in a pre-training sequence, i.e., $\mathcal{W} = [X_{u,i}, y_{u,i}]$, the model has chance to optimize the following objective function:

$$\max p(X_{u,i}, y_{u,i}) = \max \int_{\theta} p(X_{u,i}, y_{u,i}|\theta)p(\theta)d\theta. \tag{6}$$

Since $p(X_{u,i}, y_{u,i}|\theta) = p(X_{u,i}, y_{u,i}, \theta)/p(\theta)$, and $p(X_{u,i}, y_{u,i}, \theta) = p(y_{u,i}|X_{u,i}, \theta)p(X_{u,i}, \theta)$, the objective can be transformed into:

$$\max p(X_{u,i}, y_{u,i}) = \max \int_{\theta} p(y|X_{u,i}, \theta)p(X_{u,i}|\theta)p(\theta)d\theta. \tag{7}$$

We observe that Eq. (7) has the same form as Eq. (4), indicating that pre-training language models on the texts containing the content of $(X_{u,i}, y_{u,i})$ could improve recommendation performance by enabling more accurate estimation of the probabilities. However, pre-training small language models on a large and general corpus containing the interaction contexts may not be beneficial since they inadvertently allocate their limited parameters to encode irrelevant documents. In the next subsection, we introduce how to refine a

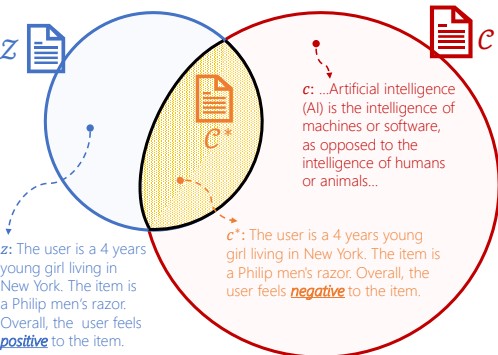

$\mathcal{Z}$

$\mathcal{C}$

$\mathcal{C}^*$

**c:** ...Artificial intelligence (AI) is the intelligence of machines or software, as opposed to the intelligence of humans or animals...

**z:** The user is a 4 years young girl living in New York. The item is a Philip men's razor. Overall, the user feels _positive_ to the item.

**c*:** The user is a 4 years young girl living in New York. The item is a Philip men's razor. Overall, the user feels _negative_ to the item.

**Figure 2: Refining corpus for model pre-training (RCMP) in system cold-start recommendation.**

general corpus into a smaller and more informative one, towards pre-training small language models for recommendation.

#### 3.3.2 Refined Corpus Extraction for Pre-training.

We consider a general corpus $C$ collected from various real-world sources, containing a recommendation-related corpus in the cold-start scenario, i.e., $C^* \subset C$. Pre-training $f_{\text{lm}}$ on general corpus $C$ benefits to cold-start recommendations. However, this approach is resource-intensive and might not yield significant gains for small language models, considering their limited parameters in storing all the information in $C$. Also, it is difficult to precisely locate and extract the $C^*$ subset to reduce the data size. Since $C^*$ is intractable in the cold-start setting, we propose to seek a refined corpus $\hat{C} \subset C$ with only $K$ documents but preserving the information of $C^*$.

Specifically, we construct $\hat{C}$ based on a self-constructed corpus $\mathcal{Z}$. Here, $\mathcal{Z}$ is the complete set of combinations of all possible user/item profiles and sentiment polarities, i.e., $\mathcal{Z} = \{[f_{\text{prompt}}(c_u, c_i), y]\}$, for $\forall c_u \in \mathbb{R}^{d_U}, \forall c_i \in \mathbb{R}^{d_I}$, and $\forall y \in \mathcal{V}_{\text{pos}} \cup \mathcal{V}_{\text{neg}}$. The relation among $\mathcal{Z}, C,$ and $C^*$ is visualized in Figure 2. We can have $C^* \subset \mathcal{Z}$ since $\mathcal{Z}$ exhausts all possibilities in the cold-start scenario. Moreover, since both the general corpus and our constructed corpus contain $C^*$, we can obtain $C^*$ by finding the subset $\hat{C}$ maximizing the mutual information between $C$ and $\mathcal{Z}$, which is formed as:

$$
\begin{aligned}
\hat{C} &= \arg\max_{C' \in C, |C'| = K} MI(C'; \mathcal{Z}) \\
&= \arg\max_{C' \in C, |C'| = K} H(\mathcal{Z}) - H(\mathcal{Z}|C') \\
&\propto \arg\max_{C' \in C, |C'| = K} -H(\mathcal{Z}|C') \\
&= \arg\max_{C' \in C, |C'| = K} \sum_{z \in \mathcal{Z}, c \in C'} p(c, z) \log p(z|c),
\end{aligned}
\tag{8}
$$

where $MI(\cdot; \cdot)$ denotes mutual information [44], $H(\cdot)$ and $H(\cdot|\cdot)$ refer to the entropy and conditional entropy, $p(\cdot, \cdot)$ and $p(\cdot|\cdot)$ are the joint and conditional probability of two pieces of texts. Through our design, we can extract documents $\hat{C}$ most correlated to the cold-start scenario, implicitly containing the related interaction records $(X_{u,i}, y_{u,i})$. Pre-training language model $f_{\text{LM}}$ on $\hat{C}$ will empower PromptRec with better in-context recommendation performance for the cold-start scenario.

Practically, the above two probabilities can be estimated with a pre-trained language model $g_{\text{LM}}$. Specifically, the joint probability $p(c, z)$ can be approximated with the similarity of two document embeddings [65]: $p(c, z) = \frac{1}{1+\exp(-e_c \cdot e_z^\top)}$, where $e_c$ as well as $e_z$ are

generated representations of documents $c$ and $z$ from $g_{\text{LM}}$. Also, the conditional probability $p(z|c)$ is estimated by [6, 36]: $\log p(z|c) = \frac{1}{|z|} \sum_{l=0}^{|z|} \log g_{\text{LM}}(z_l|c, z_1, ..., z_{l-1})$. Please note that, $g_{\text{LM}}$ could be a different language model than $f_{\text{LM}}$ used in predicting $\hat{r}_{u,i}$. The two models are applied at different stages, where $g_{\text{LM}}$ is for data pre-processing, and $f_{\text{LM}}$ is for prediction. In this section, we refine the large corpus $C$ to a small one $\hat{C}$, leading to a better in-context recommendation performance for our cold-start scenario via pre-training model $f_{\text{LM}}$ on it, which is verified by our experiments.

### 3.4 Transferable Prompt Pre-Training

#### 3.4.1 Prompt Pre-training Meets In-context Recommendation.

We start with theoretically analyzing why training prompt templates with user-item interactions is beneficial to in-context recommendations. Without loss of generality, we consider a trainable prompt template $\mathcal{S}$ as a prefix of user-item context $X_{u,i}$. Then, the learning objective of Eq. (3) is reformalized as:

$$
\min \mathcal{L}(r_{u,i}, \hat{r}_{u,i}) \rightarrow \max_{\mathcal{S}} p(y_{u,i}|\mathcal{S}, X_{u,i}).
\tag{9}
$$

In practice, we concatenate the word embedding of $\mathcal{S}$ and $X_{u,i}$ together to be fed into the language model $f_{\text{LM}}$, as shown in Figure 3. In the traditional supervised training paradigm, where we have sufficient training samples from the target dataset $\mathcal{D}_{\text{tgt}}$, the optimal prefix prompt $\mathcal{S}^*$ could be optimized with gradient-descent algorithms according to $\mathcal{S}^* = \arg\max_{\mathcal{S}} \sum_{(X_{u,i}, y_{u,i}) \in \mathcal{D}_{\text{tgt}}} p(y_{u,i}|\mathcal{S}, X_{u,i})$. However, under the cold-start setting, we cannot collect observed user-item interactions from our target datasets.

To overcome this challenge, we collect interaction records from other recommendation scenarios, called _source_ datasets $\mathcal{D}_{\text{src}} = \{\mathcal{D}^{(m)}\}_{m=1}^M$. Each source dataset $\mathcal{D}^{(m)} = (\mathcal{U}^{(m)}, \mathcal{I}^{(m)}, \mathbf{R}^{(m)})$ has a user set, an item set, an interaction matrix, and the profile features of users and items. Note that, the interaction matrix of each source dataset $\mathbf{R}^{(m)}$ is not empty, while the target matrix $\mathbf{R}_{\text{tgt}} = \emptyset$ is empty under the cold-start setting. In addition, the users and items in the target dataset are absent in the source datasets: $\mathcal{U}_{\text{tgt}} \cap \mathcal{U}_{\text{src}} = \mathcal{I}_{\text{tgt}} \cap \mathcal{I}_{\text{src}} = \emptyset$, where $\mathcal{U}_{\text{src}} = \mathcal{U}^{(1)} \cup \cdots \cup \mathcal{U}^{(m)}$ and $\mathcal{I}_{\text{src}} = \mathcal{I}^{(1)} \cup \cdots \cup \mathcal{I}^{(m)}$. Formally, we estimate the optimal prompt prefix $\mathcal{S}^*$ for the target dataset by pre-training on the source datasets, which is expressed under the HMM framework as:

$$
\begin{aligned}
\mathcal{S}^* = \arg\max_{\mathcal{S}} \sum_{m=1}^M \sum_{(X_{u,i}, y_{u,i}) \in \mathcal{D}_{\text{src}}^{(m)}} \\
\int_\theta p(y_{u,i}|\mathcal{S}, X_{u,i}, \theta) p(\mathcal{S}, X_{u,i}|\theta) p(\theta) d\theta,
\end{aligned}
\tag{10}
$$

where $(X_{u,i}, y_{u,i})$ denotes a user-item interaction record coming from one of the source datasets. By using the chain rule of probability, we could simplify the above objective function as:

$$
\begin{aligned}
\mathcal{S}^* = \arg\max_{\mathcal{S}} \sum_{m=1}^M \sum_{(X_{u,i}, y_{u,i}) \in \mathcal{D}_{\text{src}}^{(m)}} \\
\int_\theta p(\mathcal{S}|X_{u,i}, y_{u,i}, \theta) p(y_{u,i}|X_{u,i}, \theta) p(X_{u,i}|\theta) p(\theta) d\theta.
\end{aligned}
\tag{11}
$$

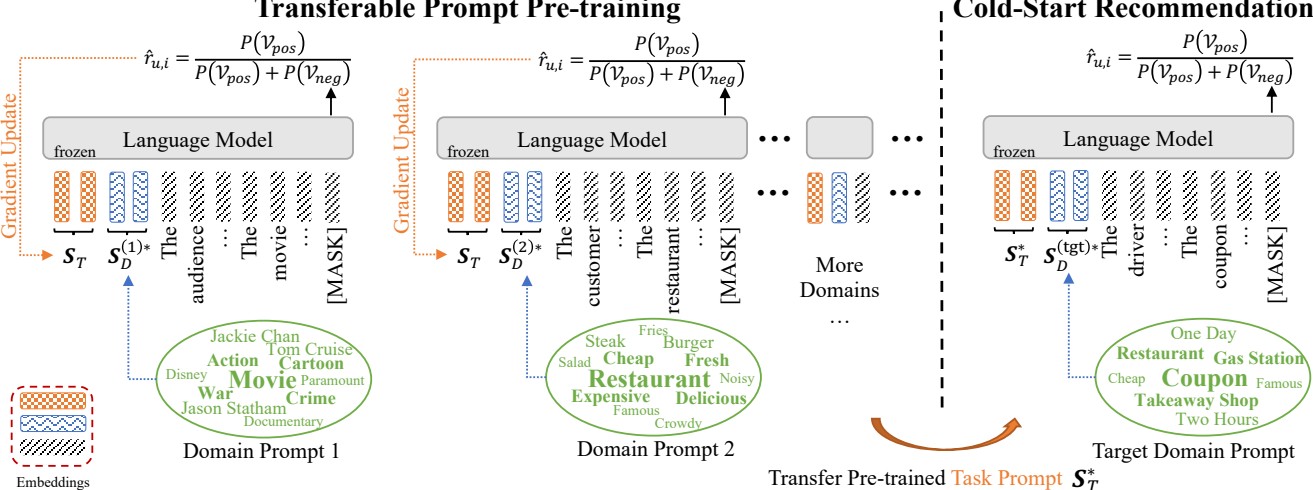

**Figure 3: Transferable prompt pre-training (TPPT) for PromptRec in system cold-start recommendation.**

Thus, the prompt posterior probability $p(\mathcal{S}|\mathcal{X}'_{u,i}, y'_{u,i}, \theta)$ indicates that the optimal prefix $\mathcal{S}^*$ captures information from three aspects: user-item contexts, sentiment words, and the mappings from user-item contexts to the sentiment words. Ideally, the information within the sentiment words, as well as the mappings, are shared across datasets, while that of user-item contexts is not. This is because different recommendation datasets usually refer to different domains, such as movies, restaurants, toys, and news. Thus, directly training on the source datasets cannot produce the optimal prefix prompt for the target domain.

### 3.4.2 Prompts Decomposition for Transferable Pre-training.

To tackle the challenge, we propose to decompose the learned prefix prompts $\mathcal{S}$ into two groups, namely *task prompt* $\mathcal{S}_T$ and *domain prompt* $\mathcal{S}_D^{(m)}$, where $\mathcal{S}_T$ describe the task "recommendation" and $\mathcal{S}_D^{(m)}$ reflects the topics of a recommendation scenario. Per discussions to Eq. (11), $\mathcal{S}_T$ should capture the information about sentiment words as well as the context-sentiment mappings, while $\mathcal{S}_D^{(m)}$ reflects unique characteristics of user-item contexts. Thus, the learning objective of our design is finally reformatted as:

$$\min \mathcal{L}(r_{u,i}, \hat{r}_{u,i}) \rightarrow \max_{\mathcal{S}_T, \mathcal{S}_D^{(tgt)}} p(y_{u,i}|\mathcal{S}_T, \mathcal{S}_D^{(tgt)}, \mathcal{X}_{u,i}), \quad (12)$$

where $\mathcal{S}_D^{(tgt)}$ is the domain prompt for the target domain $\mathcal{D}_{tgt}$. In cold-start recommendation, this objective is intractable since user-item interactions in the target domain are not available. However, by our design, a domain prompt should encode the topic within a recommendation scenario, which means it can be represented by the keywords of item profiles. In contrast, the task prompt has the ability to transfer across different scenarios, thus it should be learned from various recommendation scenarios. These insights motivate us to propose a two-stage greedy algorithm to successively optimize $\mathcal{S}_D^{(tgt)}$ and $\mathcal{S}_T$ under our cold-start setting. In the first stage, we extract the keywords of item profiles from the target domain to estimate the target-domain prompt: $\mathcal{S}_D^{(tgt)*} = g_{dom}(\mathcal{D}_{tgt})$,

where $g_{dom}$ is designed as a keyword extraction method such as TF-IDF scores. In the second stage, we leverage the interactions from source datasets to optimize the task prompt according to the reformed Eq. (11) with our prompt decomposition method:

$$\mathcal{S}_T^* = \arg\max_{\mathcal{S}_T} \sum_{m=1}^{M} \sum_{(\mathcal{X}_{u,i}, y_{u,i}) \in \mathcal{D}_{src}^{(m)}} \quad (13)$$

$$\int_\theta p(\mathcal{S}_T, \mathcal{S}_D^{(m)*}|\mathcal{X}_{u,i} y_{u,i}, \theta) p(y_{u,i}|\mathcal{X}_{u,i}, \theta) p(\mathcal{X}_{u,i}|\theta) p(\theta) d\theta,$$

where $\mathcal{S}_D^{(m)*} = g_{dom}(\mathcal{D}_{src}^{(m)})$. After obtaining $\mathcal{S}_T^*$, we could apply it to the cold-start recommendation on $\mathcal{D}_{tgt}$ as demonstrated in Figure 3. Specifically, $\mathcal{S}_T^*$ and $\mathcal{S}_D^{(tgt)*}$ are inserted as the prefix to each user-item context $\mathcal{X}_{u,i}$ in cold-start recommendations.

## 4 EXPERIMENT

We investigate three research questions (**RQ**): (1) How to evaluate the cold-start performance of PromptRec? (2) Can PromptRec make personalized recommendations in the cold-start scenario? If so, how it is sensitive to the scales of language models? (3) Can Refined Corpus Model Pre-training (Sec. 3.3) and Transferable Prompt Pre-training (Sec. 3.4) helps PromptRec generalize to small language models? To answer these questions, we introduce the first benchmark evaluating cold-start recommendation systems. We hope this benchmark can facilitate future research on developing recommendation systems under the cold-start setting.

## 4.1 Cold-start Recommendation Benchmark

Our Cold-start Recommendation Benchmark consists of three public datasets and a dataset pre-processing strategy designed to simulate the cold-start challenge in real-world scenarios. It also considers baseline methods, including traditional supervised learning methods, rule-based methods, and LLM-based methods, where all baselines and future tests will be evaluated using GAUC.

**Table 1: Statistics of datasets.**

| Dataset | #User/Feature | #Item/Feature | #Interaction | Density |
|---|---|---|---|---|
| ML-100K | 943/4 | 1682/22 | 100,000 | 3.15% |
| Coupon | 8312/12 | 6924/13 | 12,684 | 0.01% |
| Restaurant | 138/20 | 939/25 | 1,161 | 0.45% |

*4.1.1 Datasets.* The constraint of cold-start recommendation is the lack of user-item interactions during training. Under this setting, recommendation systems heavily rely on profile features to make personalized recommendations. We finally collect three datasets that meet the requirements: In-Vechical Coupon Recommendation (Coupon) [55], Mexico Restaurant Recommendation (Restaurant) [50], and MovieLens-100K (ML-100K) [17]. The Coupon dataset evaluates the performance of recommenders in delivering accurate shop discounts to drivers; the Restaurant dataset measures the ability of systems to predict user preferences for restaurants; the ML-100K dataset assesses the ability of models to recommend movies to users. Table 1 summarizes the dataset statistics.

*4.1.2 Dataset Partition.* Each dataset is split into training, validation, and test sets, where the training dataset consists of 250 samples, the valid dataset has 50 samples, and the rest of each dataset forms the test dataset. Here, the training dataset is retained only to enable the evaluation of recommendations for future works, where a small number of interaction records are available for model tuning. The validation dataset is included and kept smaller than the training dataset. All models are compared by their averaged performances on the test dataset over multiple random seeds.

*4.1.3 Data Preprocessing.* In this paper, we treat recommendation as the Click-Through Rate (CTR) prediction problem. Since the initial labels of these datasets are the intensity of user preferences toward items, we transform them into binary labels $\{0, 1\}$ by introducing a threshold [35, 67], so that they can be used as benchmarks for CTR prediction. Here, the thresholds for ML-100K, Coupon, and Restaurant datasets are 4.0, 1.0, and 2.0, respectively.

*4.1.4 Metrics.* The CTR prediction is a binary classification task that could be evaluated by the ROC-AUC score [15]. However, AUC has limitations in personalized recommendations due to its computation involving all user-item interactions, leading to unexpected interference between different users [68]. The Group-AUC (GAUC) [18, 68] score addresses it by computing AUC for each user separately and then aggregating them by weighted average:

$$GAUC = \sum_{u \in \mathcal{U}} \frac{\#history_u \times AUC(u)}{\sum_{u \in \mathcal{U}} \#history_u}, \tag{14}$$

where $\#history_u$ is the number of history records for user $u$, and $AUC(u)$ is the AUC over all interactions records for user $u$.

*4.1.5 Baseline Methods.* We consider two categories of baseline methods. The first one includes baselines that rely on human-designed rules, such as randomly recommending items to users *Random*. The second category includes LLM-based unsupervised methods, which use verbalized features of users and items as inputs and make recommendations by using outputs from LLMs without fine-tuning. For example, *EmbSim* [66] generates two embeddings

**Table 2: PromptRec for cold-start recommendation.**

| Strategy | LLM | ML-100K | Coupon | Restaurant |
|---|---|---|---|---|
| *Baselines* | | | | |
| Random | - | $50.10_{\pm 0.13}$ | $49.76_{\pm 1.33}$ | $50.44_{\pm 2.40}$ |
| EmbSim | BERT | $50.22_{\pm 0.01}$ | $50.31_{\pm 0.12}$ | $51.93_{\pm 0.87}$ |
| PairNSP | BERT | $48.88_{\pm 0.01}$ | $54.14_{\pm 0.11}$ | $47.70_{\pm 2.77}$ |
| ItemLM | BERT | $50.42_{\pm 0.01}$ | $31.98_{\pm 0.16}$ | $49.25_{\pm 1.76}$ |
| *Ours* | | | | |
| | BERT | $52.39_{\pm 0.01}$ | $\mathbf{63.77}_{\pm 0.18}$ | $\mathbf{55.49}_{\pm 1.41}$ |
| PromptRec | GPT2 | $54.45_{\pm 0.01}$ | $55.03_{\pm 0.19}$ | $51.83_{\pm 1.06}$ |
| | T5 | $\underline{56.16}_{\pm 0.01}$ | $51.11_{\pm 0.21}$ | $52.68_{\pm 0.88}$ |
| | LLaMA | $\mathbf{57.03}_{\pm 1.89}$ | $\underline{56.08}_{\pm 0.09}$ | $\underline{54.43}_{\pm 1.45}$ |

Results are converted to percentages for readability. We **bold** the best results and underline the second-best results in each dataset.

of the user and item verbalized features and predicts user-item preferences by taking the dot product of their embeddings. *PairNSP* applies the next sentence prediction task [6], where the verbalized features of the user and item are concatenated and fed into a language model to determine whether they belong to the same context. *ItemLM* [5] predicts the preference by calculating the likelihood of the item's name appearing within the user-item context.

## 4.2 PromptRec can Make Personalized Cold-Start Recommendations

*4.2.1 Experiment Designs.*

**Language language models.** We consider diverse large language models across different scales, architectures, pre-training strategies, and model sizes to show the generalization of PromptRec. Based on factors including accessibility and popularity, we choose *BERT-large-uncased* [6], *GPT-2-medium* [2], *T5-large* [38], and *LLaMA-7B* [48]. The numbers of their parameters span from 355M to 7B. We use their implementations and checkpoints from Huggingface [60].

**Prompting function designs.** Human experts design the prompting function $f_{\text{prompt}}$ for each dataset, which includes three components: templates, verbalizers, and labelers. Each template has the following parts: the user profile, the item profile, and the connection between the above profiles and the recommendation task. We consider two types of verbalizers, namely, the continuous-feature verbalizer and the discrete-feature verbalizer. The continuous-feature verbalizer breaks down the feature value range into multiple intervals, each of which is assigned with a natural language description by experts (e.g., age 72 is verbalized with word "old".). In contrast, the discrete-feature verbalizer directly returns a description for each feature value. For example, a certain user-item interaction from the ML-100K dataset is formatted as the following sentences via going through the human-designed template and verbalizer: "*The woman is a middle-aged writer living in Michigan. The Star Wars is categorized as an adventure, animated, romantic, scientific, war movie. The user is that woman, and the item is that movie. In short, the user feels* [MASK] *about the item.*" For causal language models such as LLaMA, the texts remain mostly the same, except that the last sentence is replaced with "*In short, the user's attitude towards the item is* [MASK]". The expert-designed labeler regards *positive* as the positive word, while *negative* as the negative word.

**Table 3: Sensitivity analysis of model sizes.**

| Family | #Params | ML-100K | Coupon | Restaurant |
|--------|---------|---------|--------|------------|
| | 29.1M | $52.10_{\pm0.01}$ | $50.50_{\pm0.16}$ | $49.29_{\pm0.81}$ |
| BERT | 110M | $53.55_{\pm0.01}$ | $62.96_{\pm0.18}$ | $52.67_{\pm1.28}$ |
| | 336M | $52.39_{\pm0.01}$ | $63.77_{\pm0.18}$ | $55.49_{\pm1.41}$ |

#### 4.2.2 Experiment Results.

Table 2 presents the results of the PromptRec approach with different backbone large language models on the proposed benchmark. We also report the results of several baseline zero-shot solutions on the top with BERT-large. In addition, Table 3 reports the results of BERT with different scales. Some observations are made as below.

***Baseline methods mostly fail to make personalized recommendations under the system cold-start setting.*** There is no baseline that consistently makes effective personalized recommendations on all datasets under the system cold-start setting. Some possible reasons are as follows. Essentially, personalized recommendation requires the models to capture the fine-grained differences between items from the same category. However, the two pre-training tasks (i.e., text representation and NSP [6]) of EmbSim and PairNSP can only be used to distinguish coarse-grained semantics. Thus, they can not well handle the recommendation task. On the other hand, although ItemLM relies on a fine-grained pre-training task (i.e., MLM [6]), it suffers from linguistic bias, which is discussed in Sec. 3.1. This observation shows that it is very challenging to conduct system cold-start recommendations.

***PromptRec could be generalized to various LLM families for system cold-start recommendation.*** PromptRec shows a significant improvement over the Random strategy with every LLM candidate on each dataset, indicating its strong generalization ability. Also, PromptRec improves the performance of the Random strategy on the three datasets by up to 6.93%, 14.01%, and 5.05% GAUC, respectively. This result demonstrates that large language models could make personalized recommendations with their strong in-context learning ability.

***PromptRec is sensitive to language model sizes.*** In Table 3, we observe that the BERT family generally displays a gradual improvement in performance with an increase in model sizes. Notably, the 29.1M variation cannot provide sufficient cold-start recommendations on the Coupon and Restaurant datasets. This result aligns with recent studies [59] on the relation between model sizes and in-context learning ability. It also verifies the necessity of our improvement on small language models.

### 4.3 Small Language Models are Cold-Start Recommenders with RCMP and TPPT

#### 4.3.1 Experiment Designs.

**Small language models.** We consider four small variations in the BERT family introduced by [49], including BERT-tiny, BERT-mini, BERT-small, and BERT-medium. They share the same architecture and pre-training strategies with BERT-base/large. The smallest variation, BERT-tiny, only contains 4.4M parameters with 2 transformer

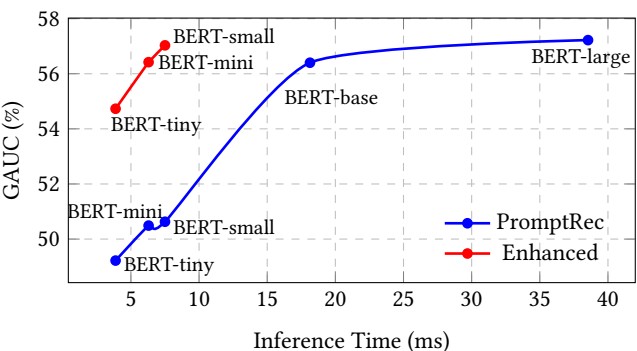

**Figure 4: Averaged cold-start recommendation performance compared with inference time over different model scales.**

layers and 128 hidden dimensions, which is the smallest publicly available general pre-trained language model.

**Details of Refined Corpus Model Pre-training (RCMP).** RCMP has two steps: refining a general corpus and further pre-training language models on the refined corpus. Specifically, we refine the general corpus according to Eq. (8) with hyper-parameter $K = 10000$. The general corpus $C$ is constructed as a subset of the C4 [37] dataset, where we randomly sample 5% documents having at least 30 unique words and 3 sentences, resulting in roughly 17 million documents. Document embeddings used for estimating $p(c, z)$ are produced by an encoder [56] trained on 1 billion paired sentences. We discovered that estimating $p(z|c)$ has minimal impact on document quality, so we treat it as a constant to expedite corpus selection. For further pre-training, we follow previous works [14], employing the Adafactor [45] optimizer with learning rate $2e^{-5}$, batch size 32, 10K training steps, and learning rate linear decay (detailed in Appendix A.1). We repeat the training over five random seeds, and report the average of the best scores for each random seed.

**Details of transferable prompt pre-training (TPPT).** For each dataset in the benchmark, we treat it as a single cold-start scenario, so we use the other two datasets to pre-train the task prompt $S_t$. We adopt a continuous task prompt, represented as trainable vectors aligned with the pre-trained model's dimension. Our training utilizes the AdamW optimizer with a learning rate of $1e^{-4}$ and a batch size of 64 for 50K steps. Performance is assessed on the validation set after each epoch. We validate the performance on the validation set at the end of each epoch. If the performance decreases consecutively twice, the learning rate will be reduced by a factor 0.5. The early stop strategy is triggered by three 3 times learning rate reduction. Domain prompts $S_d$ derive from each dataset's top TF-IDF [42] scored keywords. Every task has a 50-word task prompt, while domain prompts vary: 5, 10, 50, 100, and 200 words. Experiments for each configuration use five random seeds, with results reported for the best-performing average setup.

#### 4.3.2 Experiment Results.

Table 4 reports the results of TPPT and RCMP with the three small language models over five random seeds. Figure 4 demonstrates the relation between cold-start performance and inference time.

**Table 4: PromptRec with enhanced small language models.**

| Model | Strategy | ML-100K | Coupon | Restaurant |
|---|---|---|---|---|
| BERT-tiny (4.4M) | PromptRec | $50.72_{\pm 0.01}$ | $47.53_{\pm 0.16}$ | $49.41_{\pm 2.31}$ |
| | +RCMP | $51.15_{\pm 0.20}$ | $59.11_{\pm 0.76}$ | $50.47_{\pm 2.20}$ |
| | +TPPT | $51.86_{\pm 0.05}$ | $57.37_{\pm 0.09}$ | $53.22_{\pm 0.23}$ |
| BERT-mini (11.3M) | PromptRec | $52.45_{\pm 0.01}$ | $45.63_{\pm 0.19}$ | $53.40_{\pm 0.97}$ |
| | +RCMP | $52.77_{\pm 0.01}$ | $61.72_{\pm 3.97}$ | $53.63_{\pm 0.63}$ |
| | +TPPT | $53.17_{\pm 0.46}$ | $62.36_{\pm 0.08}$ | $53.73_{\pm 1.94}$ |
| BERT-small (29.1M) | PromptRec | $52.10_{\pm 0.01}$ | $50.50_{\pm 0.16}$ | $49.29_{\pm 0.81}$ |
| | +RCMP | $52.30_{\pm 0.20}$ | $55.25_{\pm 3.39}$ | $53.91_{\pm 1.61}$ |
| | +TPPT | $52.67_{\pm 2.75}$ | $64.57_{\pm 0.21}$ | $54.86_{\pm 0.97}$ |

*Further pre-training small language models on refined corpus enhances its cold-start recommendation performance.* In Table 4, the RCMP strategy enhances PromptRec's performance across all datasets and small language models. Notably, RCMP boosts BERT-mini's performance from 45.63% to 61.72% GAUC. For the five instances where PromptRec's performance is approximately or below 50.00% GAUC, RCMP facilitates effective in-context recommendations with the exception of BERT-tiny on the Restaurant dataset. These findings indicate that RCMP enables small language models to achieve in-context recommendations through advanced pre-training on a refined corpus.

*Pre-training transferable prompts help small language models for better cold-start recommendation performance.* In Table 4, the TPPT strategy markedly enhances PromptRec's cold-start recommendation performance across all situations. Specifically, BERT-small on the coupon dataset attains a GAUC of 64.57, surpassing the highest GAUC of 63.77 achieved by BERT-large among all candidate models. As the same observation on RCMP, TPPT facilitates in-context recommendations in all five settings where previously the language models were suboptimal. Furthermore, TPPT consistently outperforms RCMP in cold-start scenarios. These results indicate that while small language models possess the potential for in-context recommendation in cold-start situations, they require optimal prompts for activation.

*Improved small language models make effective cold-start recommendations with an efficient inference speed.* Figure 4 illustrates a trade-off between model size and performance, with BERT-large achieving the highest GAUC near 58% but with almost 40 ms inference time. However, the proposed strategies offer a significant boost of small language models from random recommendation to comparable with BERT-large in just above 5 ms.

## 5 RELATED WORK

### 5.1 Cold-start Recommendation

The phrase "cold-start" describes the situation when the recommender knows nothing about its serving objects. There are several cases of cold-start recommendations.

**New System Cold-start** Making personalized recommendations on new items to new users without any knowledge about the community the user and item came from raises the system cold-start recommendation problem [3, 8, 26, 31, 40, 43], so-called new community cold-start problem. To our best knowledge, no studies

have been conducted to formalize this issue, introduce theoretical methodologies, or establish evaluation benchmarks. Some recent studies proposed a new problem called "Zero-shot Recommendation" [7, 47], which seems the same as this case at first glance. However, a massive difference between these two concepts is that the system cold-start setting assumes the user-item interaction on the target dataset is not available during both off-line training and online inferring, while the zero-shot setting can visit the user's shopping history during the online inferring. We believe the proposed system cold-start setting is more realistic.

**New User/Item cold-start.** When a recommendation system is severing online stability, new items and users will still join day by day. Recommending these new items to users is the new item cold-start problem. Similarly, recommending items to new users causes the new user cold-start problem, and recommending new items to new users raises the new user-item cold-start problem [23, 26, 40, 43]. Different from the new system cold-start recommendation, the new user/item cold-start recommendation can train models on historical user-item interactions. Incorporating side information to enhance the quality of representing users/items is the most traditional way [13, 54, 62]. Under the assumption that side information is not available, directly mining the historical user-item interaction is more tractable [9, 23, 51]. Lately, pre-training graph neural networks is also considered as a frontier direction [16, 63].

### 5.2 In-context Recommendation

Researchers introduce in-context learning into developing recommender systems for several purposes, including interpretability [10, 24], multi-tasks learning [12, 28, 64], sequential recommendation [5, 21, 34, 47, 58, 66], and conversational recommendation [10, 57]. The intuitions behind these studies can be considered as two folds: utilizing pre-trained language models as a knowledge base to enhance recommendation [5, 28, 47, 64, 66]; using language models as a bridge to realize the interaction between humans and systems based on natural language [10, 12, 21, 24, 34, 57, 58]. These studies inspire us to push prompt learning to a more challenging scenario, the system cold-start recommendation, where the user's shopping history is not available. Moreover, in contrast with these studies that only explore *large* language models for recommendation, this study demonstrates that *small* language models could also provide cold-start recommendations under the cold-start setting.

## 6 CONCLUSIONS

This paper studies the system cold-start recommendation problem, including providing a formal definition and the first benchmark. We propose PromptRec for this problem and show that large language models can make personalized recommendations without any training samples. In addition, we provide a mathematical framework to study the behavior of language models to make in-context recommendations. We also investigate two methods to improve small language models by leveraging the datasets that are out-of-domains of the cold-start scenario. Our results demonstrate that the small language models could make personalized recommendations with their enhanced in-context ability. We encourage future research to explore the system cold-start setting for more recommendation tasks and hope they could be deployed in real-world businesses.

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

# A APPENDIX

## A.1 BERT Further Pre-training Details

We use the implementation of pre-training BERT from https://github.com/huggingface/transformers/tree/main/examples/pytorch/language-modeling to further pre-train BERT on our refined corpus. Table 5 lists some important pre-training hyper-parameters, and the hyper-parameters not listed in this table use the default settings of the source code.

**Table 5: Hyper-parameters for further pre-training BERT.**

| Hyper-parameter | Assignment |
| --- | --- |
| Training steps | 10K |
| Batch size | 32 |
| Max sequence length | 256 |
| Maximum learning rate | $2e^{-5}$ (BERT-tiny) $4e^{-5}$ (BERT-mini/small) |
| Optimizer | AdaFactor |
| Adam epsilon and beta weights | $1e^{-8}$, 0.9 and 0.95 |
| Learning rate scheduler | linear decay with warmup |
| Warmup ratio and decay rate | 500 steps and 0.1 |
| Label Smoothing Rate | 0.0 (BERT-tiny) 0.1 (BERT-mini/small) |
| Half-Precision Training | Yes |