# OpenReview forum: "Could Small Language Models Serve as Recommenders? Towards Data-centric Cold-start Recommendation"
_ACM.org/TheWebConf/2024/Conference — TheWebConf24 Oral_

### Official Review · Reviewer_e2Rp · 2023-11-16

**Novelty:** 6
**Technical Quality:** 4

**Review:**

Pros
Innovative Approach to System Cold-Start Problem: The paper presents a unique solution to the system cold-start recommendation challenge, a crucial issue in scenarios without any historical user-item interactions.
Effective Enhancement of Small Language Models: Demonstrates the potential of small language models to reach performances similar to their larger counterparts, while notably reducing inference time and resource demands.
Cons
Limitations in Addressing Transfer Learning Techniques: The paper does not fully explore the limitations of existing transfer learning techniques in the context of system cold-start recommendations.
Baseline Comparisons: The chosen baselines for comparison are relatively weak, akin to random guessing. The inclusion of stronger baselines, such as few-shot learning with randomly generated examples which could not exist in $R_{tgt}$, would offer a more robust evaluation.

**Questions:**

Prompt Design Clarification: Could you clarify the design process of the function $g_{dom}(\dot)$in the transferable pre-trained task prompt? Is it merely a selection of tokens with the highest TF-IDF scores? Additionally, if $D_{tgt}$ is accessible for determining $S_{D}^{(tgt)*}$, does this still align with the definition of a systematic cold-start scenario?
Comparison with Few-shot Learning: How does the in-context learning (ICL) approach in PromptRec compare with few-shot learning methods that use generated examples? Could you elaborate on the superiority of the Transferable Prompt approach in this context?

**Reviewer Confidence:**

3: The reviewer is confident but not certain that the evaluation is correct

**Scope:**

3: The work is somewhat relevant to the Web and to the track, and is of narrow interest to a sub-community

---

### Official Review · Reviewer_LSbh · 2023-11-20

**Novelty:** 6
**Technical Quality:** 5

**Review:**

The paper aims to solve the system cold start problem with language models. The  main challenges of the problem lie in that there is not samples for training and the inference time of large models is time-consuming. To tackle those challenges, the authors propose a data-centric method to finetune an extreme small language model. The method includes a high-quality data selecting strategy and a transferable prompt pretraining stage. The data selecting strategy aims to select recommendation-related samples from general corups and the prompt pretraining technique is designed to guide the LM to adapt to recommendation task. Experiments are then conducted on three small datasets to demonstration the effectiveness of proposed techniques. But the results are not convincing due to the insufficient baselines and the limited performance.

### Strengths
1. The writing of the paper is fluent and easy to follow.
2. The problem to be solved in this paper does exist and is of great research value in the field of recommendation systems. Most existing LM-based methods have not considered the feasibility of grounding, which are time-consuming in inference step. But this paper aims to use extremely small LM for the cold start task.
3. The data-centric method is novel, which leverages the idea of high-quality samples construction and  prompt tuning. And it provides a simple method for the hard system cold-start problem.


### Weaknesses
1. The experiments are not convincing. The selected baselines and datasets are not sufficient to demonstrate the effectiveness of the proposed method.
2. The effectiveness of the proposed method is not promising. Most reported results in experiments show very limited improvements over the Random strategy, even for such small datasets.

**Questions:**

1. Why is only one word selected as sentiment word in implementation?

2. Is there any evidence that “estimating p(z|c) has minimal impact on document quality“?

3. Since some samples remain for training, why not use them to train traditional recommendation methods, such as MF, FM, SASRec, et al? I observe that some methods are implemented in your codes, such as NeuMF, PNN, Wide&Deep, LightGCN.

4. Why are all the selected datasets so small? Do you have confidence that the method still effective in larger datasets?

**Reviewer Confidence:**

3: The reviewer is confident but not certain that the evaluation is correct

**Scope:**

4: The work is relevant to the Web and to the track, and is of broad interest to the community

---

### Official Review · Reviewer_y28L · 2023-11-21

**Novelty:** 6
**Technical Quality:** 6

**Review:**

This paper proposes a Prompt-Rec, a novel small language model-based recommender system that has the ability to generate cold-start recommendations as LLMs while significantly reducing the inference time.

This paper has the following advantages that I appreciate very much.

+ The studied problem, i.e., how to develop a model with good knowledge and reasoning ability as LLM for cold-start recommendation while reducing the inference cost to make it practical for implementation, is very important.
+ The theoretical analysis of bridging in-context learning and recommendation seems sound and interesting.
+ Experiments on multiple public datasets and the established cold-start recommendation dataset seem to demonstrate the effectiveness of the proposed method.

In addition, the authors have released the codes, which is definitely a plus.

However, I have the following questions for the authors, which should be carefully addressed before publication.

- First, the proposed Prompt-Rec needs "a self-constructed corpus" with cross-domain user-item interactions. Would these datasets be available in real-world cases? And since user/item profiles and interactions need to be transformed into textual data, would the diversity of natural language jeopardize the accuracy of such natural language representations?

- In addition, it would be nice if the authors could analyze how the "two-stage greedy algorithm" for prompt decomposition deviates from the true optima.

- Finally, I'm curious that for GPT and LLaMA, where no "small" versions of a few million parameters are available, how to use these models via the Prompt-Rec?

**Questions:**

Please refer to my summary of questions in the main review.

**Reviewer Confidence:**

3: The reviewer is confident but not certain that the evaluation is correct

**Scope:**

4: The work is relevant to the Web and to the track, and is of broad interest to the community

---

### Official Review · Reviewer_8Je8 · 2023-12-05

**Novelty:** 6
**Technical Quality:** 5

**Review:**

This paper considers the _system_ cold start setting, where a new recommender system is deployed without any previous user-item interactions being available. The system must instead be trained using _different_ recommendation tasks, with the goal being that this could generalize to the new task of interest. The approach appears novel and interesting.

The authors’ key contribution is to ask whether an LLM-based approach can help solve this problem. Moreover, they focus on using LLMs that are practical at scale in production systems, hence small language models. They first assess how models of different sizes (from 4.4M to 7B parameters) perform on this task, tested on three different “test” datasets. They find that larger models perform better, then show how to do additional training on small models to make them perform almost as well as much larger models, but be small enough to be efficient at scale. It is a very nice idea.

It is also great to see that the authors are making their code and datasets available for download, this makes it possible for readers who wish to fully understand the methodology to fully reproduce the results – although I did not test it.

A few suggestions:
* Equation (8) could be made more readable, as especially the exponentiation is unclear.
* Typo in 4.1.1: Vechical -> Vehical (matches the original citation [55]) or Vehicle (correct spelling)

**Questions:**

* What would “oracle” performance be on these tasks? That is, given the interaction data that is available, how well could a system that is not cold-start perform in theory? Understanding this would show how far the (essentially) transfer learning approach described here goes towards resolving the cold start challenge. This would also place the performance numbers in context: It is difficult to know if 57% performance is any use in practice, even if it is better than random baselines.
  * A related question is – one could then ask how much interaction data would be needed to achieve equivalent performance? I.e. the approach is useful when the system has *no* interaction data. What if it had a little, as would be collected by a practical system over time? Does the method described here help for a long time, or would it become irrelevant as soon as the practical system has a small amount of real user interaction data?
* How exactly were the datasets split in 4.1.2, to ensure no leakage between training, validation and test datasets?
* The GAUC metric gives more weights to users with a longer history. This is unusual, as it does not predict performance for a “random new user”. It would be great to better understand why this weighting is desirable.
* Figure 4 shows BERT-tiny reaching performance around 54.5%. Does this correspond to the average of the 3rd row of Table 4? It would be helpful for the numbers behind Figure 4 to be explicit somewhere in the paper.

**Reviewer Confidence:**

3: The reviewer is confident but not certain that the evaluation is correct

**Scope:**

3: The work is somewhat relevant to the Web and to the track, and is of narrow interest to a sub-community

---

### Decision · Program_Chairs · 2024-01-22

**Decision:**

Accept (Oral)

**Comment:**

This paper studies the problem of system cold start, where a new recommender does not have access to any existing item recommendations. The authors study how LLMs can help solve this problem by generalizing from other recommendation tasks.

 The reviewers recognized a number of strengths:
 * The problem of leveraging language models for recommendation is current and well presented
 * The focus on small language models - to be practical - is nice.
 * There is an interesting and relevant theoretical analysis

 As well as weaknesses:
 * The experimental protocol & how datasets are used is not described in enough detail
 * There is potential for stronger baselines
 * It would be particularly interesting what the gap is, with regards to oracle or two-stage greedy performance.

 The authors addressed the weaknesses in the rebuttal period, with most of the responses clarifying the content in the paper (although some new results were also presented, which to integrate will take some space away from other things already in the paper).

 Overall, the reviewers agree that the paper is ready for acceptance at TheWebConf, although it will also benefit greatly from the authors addressing the questions raised in the reviewers, and incorporating the improvements described.